# Biodiversity and Safety Assessment of Half-Century Preserved Natural Starter Cultures for Pecorino Romano PDO Cheese

**DOI:** 10.3390/microorganisms9071363

**Published:** 2021-06-23

**Authors:** Luigi Chessa, Antonio Paba, Elisabetta Daga, Ilaria Dupré, Roberta Comunian

**Affiliations:** Agris Sardegna, Agenzia Regionale per la Ricerca in Agricoltura, Associated Member of the JRU MIRRI-IT, Loc. Bonassai, SS291 km 18.600, 07100 Sassari, Italy; apaba@agrisricerca.it (A.P.); edaga@agrisricerca.it (E.D.); idupre@agrisricerca.it (I.D.); rcomunian@agrisricerca.it (R.C.)

**Keywords:** *Scotta-innesto*, natural starter cultures, Pecorino Romano PDO, autochthonous cultures, ex situ biodiversity preservation, microbial collections, antibiotic resistance

## Abstract

The use of biodiverse autochthonous natural starter cultures to produce typical and PDO cheeses contributes to establishing a link between products and territory of production, which commercial starters, constituted by few species and strains, are not able to. The purpose of this work was the assessment of biodiversity, at strain level, and safety of natural *scotta-innesto* cultures whose use is mandatory for the Pecorino Romano PDO cheese manufacturing, according to its product specification. The biodiversity of three *scotta-innesto*, collected in the 1960s and preserved in lyophilised form, was assessed by molecular biotyping using both PFGE and (GTG)_5_ rep-PCR profiling on 209 isolates belonging to *Streptococcus thermophilus* (30), *Lactobacillus delbrueckii* subsp. *lactis* (72), *Enterococcus faecium* (87), and *Limosilactobacillus reuteri* (20), revealing high biodiversity, at the strain level, in the cultures. The cultures’ safety was proved through a new approach assessing phenotypic and molecular antibiotic resistance of the cultures *in toto*, instead of single strains, while the safety of *Enterococcus faecium* isolates was investigated according to EFSA guidelines. The use of natural biodiverse cultures for the production of microbial starters for typical and PDO cheeses, such as Pecorino Romano, could be an opportunity for recovering the cheese microbiota biodiversity lost during years of commercial starters use.

## 1. Introduction

The Pecorino Romano PDO is one of the most exported Italian cheeses in the world and, with more than 3000 tons per year being produced, it represents about 20% of total *pecorino* cheese [1,2,3]. It is mainly manufactured in Sardinia (Italy) [4], but it can be produced also in Latium and in the Province of Grosseto (Tuscany). Pecorino Romano is a hard cheese that can be made from raw or thermised ewes’ milk, usually inoculated with a traditional natural starter culture, the *scotta-innesto*. Operatively, *scotta-innesto* has usually been prepared by inoculating *scotta* (the residual whey obtained from ricotta cheese manufacturing) with an aliquot of the stock *scotta-innesto* culture obtained from previous Pecorino Romano manufacturing the day before, and incubated overnight until acidification. However, in recent decades, the technological innovations and the improvement of hygienic conditions of milking and manufacturing processes, requested by the EU regulation [5], have resulted in a depletion of useful lactic acid microflora in the dairy environment, thus affecting the correct milk acidification and allowing the development of detrimental microorganisms. According to the cheese product specification [6,7], to overcome this problem, the *scotta* can be integrated also with commercial starter cultures consisting of few selected strains of lactic acid bacteria (LAB), such as *Streptococcus thermophilus*, *Lactobacillus delbrueckii*, *Lactobacillus helveticus*, certified as autochthonous [8]. This practice, in the long term, could lead to the replacement of natural microbial communities colonising the dairy plants, with few selected strains, to the consequent detriment of the useful local microbial biodiversity, which is considered able to improve the sensory richness of artisanal and typical cheeses [9,10].

The present study, following up the work recently published by Chessa et al. [11,12], aims to recover the eroded biodiversity by reintroducing in dairy plants half-century preserved lyophilised cultures, which are strongly linked to the territory and which reflect the production environment of the past, still uncontaminated by commercial selected strains, in line with the product specification of the Pecorino Romano PDO cheese. Therefore, the biodiversity, at strain level, of three autochthonous natural *scotta-innesto* (SR30, SR56, and SR63), belonging to the BNSS Agris Sardegna microbial collection, was investigated. In addition to their identification, the characterisation of microbial strains used in food production includes the evaluation of aspects related to their safety [13]. However, at the moment, though increasing attention has recently been focused on natural cultures, consisting of an indefinite number of strains, no mandatory requirements are provided for their safety assessment. Thus, in this study, a new step-by-step approach, applying similar criteria to those used for selected strains, was adopted for the natural *scotta-innesto* cultures *in toto*. Moreover, special attention was paid to the *Enterococcus faecium*, a species present in natural *scotta-innesto*, not included in the QPS list due to its potential pathogenicity, but which plays an important role in ripening and sensory connotation of the cheese.

## 2. Materials and Methods

### 2.1. Experimental Design

The Agris Sardegna BNSS microbial collection (http://www.mbds.it, accessed on 21 June 2021) includes, along with over 10,000 isolates, about 80 undefined natural cultures in *scotta* harvested from 9 Sardinian cheese factories producing Pecorino Romano DOP and stored in freeze-dried form since the late 1960s. Three of these cultures, collected *in toto* from two dairy plants (indicated as A and B) located in Berchidda (Sardinia, Italy) (SR30 from A, SR56 and SR63 from B), were reactivated in reconstituted powder *scotta*. In particular, the cultures *in toto* and their isolates were investigated for antibiotic resistance and evaluated for their microbial biodiversity at strain-level.

### 2.2. Reactivation of the Cultures

The three natural biodiverse starter cultures SR30, SR56, and SR63 were reactivated in cow skim milk supplemented with 0.5% of yeast extract and incubated at 42 °C overnight. Then, the reactivated cultures were inoculated in sterile *scotta* reconstituted in water (72.2 g/L *scotta* purchased from Alimwhey, Alimenta S.r.l., Cagliari, Italy) and incubated overnight at 42 °C.

### 2.3. Microbial Counts and Isolation

Microbial plate counts for the enumeration of the main LAB groups constituting the natural cultures SR30, SR56, and SR63 were performed in: M17 agar (Microbiol, Cagliari, Italy) at 45 °C for 72 h aerobically for thermophilic cocci; MRS agar pH 5.4 (Microbiol) at 45 °C for 48 h anaerobically for thermophilic lactobacilli; FH agar [14] at 37 °C for 72 h anaerobically, for mesophilic lactobacilli; KAA (Microbiol) at 42 °C for 18–24 h aerobically for enterococci; mannitol salt agar (MSA) at 30 °C for 72 h in aerobiosis for staphylococci; VRBA mug at 37 °C for 18 h aerobically for coliforms. For anaerobic conditions, OxoidTM AnaeroGenTM (Thermo Fisher Scientific, Waltham, MA, USA) was used. Microbial counts were performed in triplicate and the results expressed as average values ± standard deviation (SD) Log CFU/mL.

Up to 10 colonies were picked up from each medium seeded for microbial counts of each culture. Isolates were frozen at −80 °C in the appropriate culture medium with 15% glycerol until purification and identification.

### 2.4. Molecular Identification and Biotyping of the SR Isolates

Purified cocci and bacilli isolates from SR30, SR56, and SR63 were grown overnight, in M17 or MRS media, respectively, and then molecular identification at species level was performed by species-specific PCR using primers listed in the Appendix A and PCR protocols described above, as well as strain-typing by (GTG)_5_ rep-PCR and pulse field gel electrophoresis (PFGE) profiling.

The microbial fingerprinting for *Streptococcus thermophilus*, *Lactobacillus delbrueckii* subps. *lactis*, *Limosilactobacillus reuteri*, and *Enterococcus faecium* was performed by (GTG)_5_ rep-PCR analysis, as described by Chessa et al. [12] using an FTA^®^ Disc for DNA analysis (GE Healthcare) directly as a template. PCR products were separated on agarose gel (1.8% *w*/*v*), at 90 V (222 V/h) in Tris-acetate buffer, stained in ethidium bromide (0.5 μg/mL).

Molecular typing of the isolates was performed with PFGE, following different protocols depending on the bacterial species investigated (*Streptococcus thermophilus*, Lac*tobacillus delbrueckii* subps. *lactis*, *Enterococcus faecium*, and *Limosilactobacillus reuteri*). For *S. thermophilus*, DNA extraction was performed according to Tosi et al. [15] using 20 U of the restriction enzyme *Sma*I for 4 h, whereas for *E. faecium,* the protocol described by Graves and Swaminathan [16] was followed, using 25 U of *Sma*I for 4 h. PFGE was conducted, with CHEF-Mapper PFGE (Bio-Rad Laboratories, Hercules, CA, USA), using the running protocol described by the same authors, with some modifications. The running time of a total 16 h was divided into 3 blocks: block 1 of 5 h, initial switch time 1 s final switch time 20 s; block 2 of 5 h, initial switch time 1 s final switch time 5 s; block 3 for 6 h, initial switch time 10 s final switch time 40 s. For *L. delbrueckii* subsp. *lactis*, DNA extraction was carried out according to Gosiewski and Brzychczy-Wloch [17], using *Xba*I (25 U for O/N incubation) restriction enzyme. Electrophoresis was conducted in 1.5% agarose gel at 6 V/cm, with 120° angle, for a total of 20 h at 13 °C, with the following modified running parameters (3 blocks): (i) 5 h, initial time 1 s final time 5 s; (ii) 9 h, initial time 0.5 s final time 3 s; (iii) for 6 h, initial time 0.1 s final time 1 s. For *L. reuteri*, DNA was extracted following the same protocol adopted for Lac*tobacillus delbrueckii* subps. *lactis*, whereas electrophoresis was conducted in 1% agarose gel, following the protocol used for *E. faecium*.

Gel images of both (GTG)_5_ rep-PCR and PFGE gels were acquired with the UV transilluminator FireReader V4 (UVITec, Warwickshire, UK) in tiff format, then elaborated for cluster analysis by BioNumerics (v. 6.6.11; Applied Maths, Sint-Martens-Latem, Belgium) using Pearson correlation indexes and the unweighted pair group method using arithmetic averages (UPGMA). For most of the isolates, a composite data set comparison tool was also used analysing both molecular fingerprints, while several isolates that gave no suitable profiles with the PFGE technique were typed using only the (GTG)_5_ rep-PCR profile.

### 2.5. Evaluation of Phenotypic Antibiotic Resistance of the Starter Cultures

The antibiotic resistance of the natural cultures SR30, SR56, and SR63, *in toto*, was performed in the broth media ISTL (IST + lactose 10 g/L), for cocci, and LSM, for lactobacilli, supplemented with different concentrations of ampicillin (Amp), chloramphenicol (Chl) clindamycin (Cli), erythromycin (Ery), gentamycin (Gen), tetracycline (Tet), or tylosin (Tyl), according to the ISO 10932:2010 [IDF 223:2010] [18]. The antibiotic concentrations tested, chosen based on the indications provided by the FEEDAP EFSA Panel in the Scientific Opinion “Guidance on the characterisation of microorganisms used as feed additives or as production organisms” [13], by M100 S30 of Clinical and Laboratory Standards Institute (CLSI) [19] or The National Antimicrobial Resistance Monitoring System (NARMS) [20], were reported in Table 1. Operatively, 0.1 mL of one 10-fold dilution in sterile physiological solution of SR30, SR56, or SR63, was inoculated in 10 mL of sterile ISTL or LSM broth, supplemented with one of each of the abovementioned antibiotics and incubated at 37 °C overnight. Antibiotic tolerance was evaluated by visual examination of microbial growth. Three replicates for each culture/medium/antibiotic/concentration were performed.

Moreover, the concentration of antibiotic-resistant bacteria present in SR56 was estimated in ISTL or LSM media supplemented with Tet (4, 8, and 16 mg/mL) or Ery (2, 4, and 8 mg/mL). Operatively, 0.1 mL of three 10-fold serial dilution in physiological solution was inoculated in 10 mL fresh broth and then incubated overnight at 37 °C. The highest cultured dilution yielding growth allowed the estimation of Log cells/mL. Furthermore, each culture was microscopically inspected for cell morphology characterisation.

### 2.6. Molecular Analysis for the Safety Evaluation and Identification of SR56 Isolates

For SR56, PCR both for species identification and detection of the antibiotic resistance genes (ARGs) *tet*M, *tet*L, *tet*S, *tet*W, *erm*A, and *erm*B, were performed on the culture *in toto* and microbial isolates. DNA was extracted using microwave oven treatment [21] from 1 mL of SR56 grown in ISTL or LSM broth media supplemented with Tet (4, 8, and 16 mg/mL) or Ery (2, 4, and 8 mg/mL), and 0.1 mL of the same broth cultures were spread onto M17 agar medium. A total of 49 colonies were picked up, and the DNA was extracted as reported above. PCR for the identification at genus (*Enteroccoccus* spp.) or species (*E. faecium*, *E. faecalis*, *E. durans*, and *S. thermophilus*) level, and for the detection of the ARGs *tet*M, *tet*L, *tet*S, *tet*W, *erm*A, and *erm*B were carried out in a total volume of 25 µL consisting of 22 µL MegaMix (Gel Company, Inc., San Francisco, CA, USA), 1 µL of each primer 50 mM, and 1 µL of DNA sample, by Eppendorf Mastercycler (Eppendorf, Hamburg, Germany) with the following protocol: 7 min of denaturation at 94 °C, followed by 30 cycles of 30 s at 94 °C, 30 s at the specific temperature of each primer set (Appendix A), 30 s at 74 °C, and final extension at 72 °C for 7 min. PCR amplicons were separated by electrophoresis assay on 1.8 % (*w*/*v*) agarose gel in Tris-acetate buffer, stained in ethidium bromide solution (0.5 µg/mL) and observed by UV transilluminator.

For the assessment of the safety of the three cultures (SR30, SR56, and SR63) *in toto* and the *E. faecium* isolated from them, the ampicillin minimal inhibitory concentration (MIC) (for one representative isolate for each biotype) was determined and the detection of pathogenic-related genes *hyl*Efm, *esp*, and IS*16* (for all the isolates, 87) was performed according to EFSA guidelines [22]. PCR reactions were carried out using the same protocol described above, with primers and annealing temperatures specific for each gene target (Appendix A).

## 3. Results and Discussion

### 3.1. Microbial Counts, Molecular Identification and Biotyping of the SR Isolates

Total viable bacteria, thermophilic cocci and bacilli, heterofermentative lactobacilli, enterococci, staphylococci and coliforms for the three cultures SR30, SR56, and SR63 for Pecorino Romano PDO manufacturing, collected between 1968 and 1970 and preserved *in toto* in freeze-dried form, were enumerated. Plate counts in the chosen elective media were reported in Figure 1. SR30 was composed exclusively by cocci shaped bacteria (8.72 Log CFU/mL in M17 agar medium and 6.32 Log CFU/mL in KAA medium), whereas SR56 and SR63 were composed mainly of thermophilic lactobacilli (9.41 and 8.67 Log /CFU/mL, respectively) and enterococci (8.03 and 2.56 Log CFU/mL). Moreover, also heterofermentative lactobacilli were found in SR63 (4.18 Log CFU/mL), whereas no staphylococci or coliforms were found in the three cultures. The composition of the natural SR cultures revealed biodiversity in terms of microbial groups useful for Pecorino Romano cheese manufacturing.

A total of 209 isolates from the natural cultures were molecularly identified at species level: 30 *S. thermophilus* and 45 *E. faecium* were isolated from SR30. Both cocci-shaped bacteria (*E. faecium*, 24 and 18) and homofermentative thermophilic lactobacilli (*L. delbrueckii* subsp. *lactis*, 40 and 32) were isolated from SR56 and SR63, respectively, and 20 *Limosilactobacillus reuteri* were isolated from SR63. Thermophilic lactic acid bacteria (LAB), such as *S. thermophilus* and *L. delbrueckii* subsp. *lactis*, starter LAB (SLAB) predominating the early phases of acidification of *scotta-innesto*, cheese-making, and cheese ripening [23], are able to produce a high level of lactic acid [24] and generally dominate the microflora of the natural starter *scotta-innesto,* along with *Lactobacillus helveticus* (the latter, together with *L. delbrueckii* subsp. *bulgaricus*, is most frequently isolated from *scotta-innesto* used in Lazio) [25,26]. They are found also during ripening of Pecorino Romano PDO cheese together with non-starter LAB (NSLAB). Among these, mesophilic facultative heterofermentative lactobacilli (such as *Lactiplantibacillus plantarum* and *Lacticaseibacillus casei*), thermophilic obligate heterofermentative lactobacilli (such as *Limosilactobacillus fermentum*) and enterococci (such as *E. faecium* and *E. durans*) play an important role in cheese maturation, conferring typical flavours and aromas, through processes of proteolysis, lipolysis and/or metabolism of citrate. In addition, some strains can be able to produce bacteriocines, or are known to have probiotic activity [21,27,28,29,30]. As previously reported by Chessa et al. [11,12], in *scotta-innesto* for Pecorino Romano cheese, *L. reuteri*—an obligatory heterofermentative LAB able to follow metabolic pathways leading the development of aromas precursors enhancing the sensory quality of the final product [31], which can have probiotic properties with beneficial effects on human health [32,33,34,35]—was also found. Moreover, *L. reuteri* can also contribute to the safety of dairy products by producing antimicrobial substances, such as reuterin, able to inhibit growth of several pathogens such as *Listeria monocytogenes*, *Escherichia coli* O157:H7 (EHEC) [36], *Staphylococcus aureus*, *Salmonella choleraesuis* ssp. *choleraesuis*, *Yersinia enterocolitica*, *Aeromonas hydrophila* ssp. *hydrophila* and *Campylobacter jejuni* [37]. Enterococci are NSLAB involved in the development of sensory richness in several traditional cheeses of the Mediterranean area, exerting proteolytic and lipolytic activity, and citrate metabolism [28]. However, their role in fermented foods is controversial, since some enterococci—in particular *E. faecium*, ubiquitous of dairy products and gastrointestinal tract of humans and animals [38]—are often resistant to antibiotics and potential human pathogens [28,39,40]. The secondary adventitious microflora belonging to the genera *Staphylococcus*, which is useful for conferring typical characteristics to the cheese [29], was not found in the cultures investigated in this study.

Molecular biotyping was carried out using (GTG)_5_ rep-PCR and PFGE (with specific restriction enzymes for each microbial group), then composite analysis including both methods was performed. The composite analysis (93% similarity cut-off) carried out on the 30 *S. thermophilus* isolates from SR30 revealed five different biotypes (Figure 2). Four *S. thermophilus* isolates were singletons, whereas most of the isolates were grouped into a main single cluster (<93.1% similarity), which in turn consisted of three subclusters with high similarity within each cluster (<95.0%, <96.1%, and 97.6%, respectively) (Figure 2).

The molecular biotyping performed on 72 *L. delbrueckii* subsp. *lactis* isolates revealed, with 93% similarity cut-off, 30 different biotypes isolated from SR56 and SR63 natural starter cultures (Figure 3). In particular, 54 of the *L. delbrueckii* subsp. *lactis* isolates were distributed into 12 clusters, plus 18 singletons. The SR56 and SR63 isolates often clustered together, therefore no culture-dependent clustering was observed (Figure 3). Indeed, the two cultures came from the same dairy plant (B).

Among the 87 *E. faecium*, isolated from all the three cultures SR30, SR56, and SR63, 68 were typed by analysing the (GTG)_5_ rep-PCR and PFGE profiles by Bionumerics composite data set tool, which found, with 93% similarity cut-off, 35 different biotypes, showing a culture-dependent clustering (Figure 4). For SR30 eight biotypes, out of the 12 found, clustered apart from SR56 and SR63 isolates (sharing 50% similarity with them), while four were singletons and shared similarity ≤ 72.1% with isolates from the other two cultures. Additionally, for SR56 and SR63, a culture-dependent clustering was observed: 20 *E. faecium* biotypes were found for SR56 (16 were singletons) whereas four were found for SR63. For 19 (11 from SR30 and 8 from SR63) out of the 87 isolates, no restriction profile was obtained by *Sma*I enzyme. Therefore, they were typed only by the (GTG)_5_ rep-PCR technique, obtaining six biotypes: three including isolates coming only from SR30, two from SR63, and one from both SR30 and SR63 (Figure 5). Among the 20 *L. reuteri* isolates, found only in SR63 and characterised by composite analysis, eight different biotypes were found (Figure 6). The isolates were grouped into six clusters (≤93% similarity within each cluster) plus two singletons.

Overall, the natural starter cultures investigated in this study were composed of four species, each made up of at least five different biotypes. In particular, five biotypes for *S. thermophilus*, eight for *L. reuteri*, and up to 30 and 35 biotypes for *L. delbrueckii* subsp. *lactis* and *E. faecium*, respectively, were found. The cut-off used in this study (93% similarity) for the definition of biotypes was calculated based on the comparison of three independent (GTG)_5_ rep-fingerprints of 12 strains. A similarity threshold of 93.1%, consistent with that reported by Gevers et al. [41], was determined (data not shown) using Pearson correlation coefficient, which considers both presence/absence and weight of bands in profiles. In contrast, Dice takes into account only the bands’ presence/absence, so this is the most suitable correlation coefficient to be applied for PFGE profiles analysis. However, since in this study composite analysis using two molecular techniques—one based on PCR amplification (i.e., (GTG)_5_ rep-PCR) and the other on restriction reaction (PFGE)—was applied, Pearson correlation coefficient should be used. Therefore, three independent PFGE profiles of the same strain were analysed, and the similarity threshold calculated was 94.7% (data not shown). When PFGE profiles alone were analysed, using the Dice coefficient, a 100% similarity threshold was obtained, allowing a greater number of biotypes to be distinguished. Nevertheless, in order to not overestimate the biodiversity, the lowest cut-off calculated (93%) was used for the composite analysis. Thus, a greater genetic variability could be discovered if more stringent criteria are used to perform strain profiles analysis or culture-independent molecular techniques are applied, since it was estimated that approximately only 1% of microbial biodiversity can be detected using culture-dependent techniques [9].

### 3.2. Antibiotic Resistance in the Starter Cultures and Safety Assessment of Enterococcus Faecium Isolates

In this study, a novel approach was followed to assess the presence of antibiotic-resistant bacteria in the three natural *scotta-innesto*. Indeed, as a general rule, antibiotic resistance protocol tests reported in the literature are established for pure cultures of well-identified single strains, but, at the beginning of this trial, no bacteria isolation was performed, and the composition in species of the natural starter cultures was unknown. However, since the lack of antibiotic resistance was a prerequisite to go ahead in the study of the mixed natural cultures, and, on the basis of microbial plate counts, they were supposed to be made up of species with different cut-off resistance values, developing a new strategy became crucial. Therefore, a step-by-step approach, where each step was performed based on the result obtained in the previous one, was adopted. The mixed cultures *in toto* were inoculated in two broth media, suitable for cocci or bacilli antibiotic resistance assessment, containing antibiotics at different concentrations. The seven antibiotics tested and their concentrations were chosen according to the indications provided by the FEEDAP EFSA, CLSI, and NARMS [13,19,20] for *E. faecium*, a NSLAB species commonly found in ewe milk and cheese, with the highest cut-off among the microbial groups present in the cultures. Furthermore, it is considered a potential foodborne pathogen and shows a higher level of resistance compared to the other microbial species commonly found in dairy microbial consortia [42].

The phenotypic assessment revealed microbial growth in ISTL and LSM media for SR30 and SR63, but only when supplemented with antibiotics at concentrations consistent with the cut-off established by EFSA [13], CLSI [19], or NARMS [20] for *E. faecium* (Table 1). Therefore, there was no reason to believe that bacteria with acquired resistance were present in these cultures. However, the microbial consortium of SR56 was able to grow in media supplemented with clindamycin, gentamycin, tetracycline, and erythromycin above the EFSA cut-offs [13] (Table 1). Although clindamycin EFSA [13] cut-off for *E. faecium* is 4 mg/L, the intrinsic resistance of enterococci to this antibiotic is reported in the literature [43], and no cut-off is indicated by CLSI [19] for this genus. Gentamicin resistance found at the cut-off (32 mg/L) stated by EFSA [13] should not be considered a cause of concern, since a high cut-off (>500 mg/L) is reported by NARMS [20] for enterococci to be considered as resistant, and a combination of multiple antibiotics is often needed in clinical infections treatment [39,44,45]. Regarding the tetracycline and erythromycin resistance, the second step was to estimate the concentration of antibiotic-resistant bacteria present in the SR56 culture. Tetracycline resistance at 8 mg/L was observed in media for both cocci and bacilli (ISTL and LSM, respectively) (Table 2). However, microscopic observation of positive samples revealed an overwhelming prevalence of coccal forms, while the very rare cells with bacillary morphology, present only in the lowest dilution tested, were most likely those that were inoculated, unable to reproduce in antibiotic-supplemented media and therefore not resistant. The presence of cocci potentially resistant to 8 mg/L of tetracycline was estimated in the order of 4 Log CFU/mL, whereas the concentration of potentially resistant bacilli was estimated in around 2 Log CFU/mL (Table 2). Similarly to what was observed for tetracycline, for erythromycin, different responses of the SR56 culture in the two media were observed (Table 2). In particular, only cocci were found, though in LSM medium, at all the erythromycin concentrations tested (2, 4, and 8 mg/L, corresponding to *S. thermophilus* EFSA cut-off, *E. faecium* EFSA cut-off, and *E. faecium* CLSI cut-off, respectively) [13,19] (Table 2). Microbial growth was observed up to the highest microbial decimal dilution (1/1000) inoculated, when 2 or 4 mg/L erythromycin were present, while at 8 mg/L of erythromycin, growth was found at one dilution lower (1/100) (Table 2). Surprisingly, in ISTL, the elective medium for cocci-shaped bacteria, microbial growth was detected only at the lowest decimal dilution tested (1/10). Therefore, even considering the EFSA cut-off instead of the CLSI one, *E. faecium* able to grow at 8 mg/L erythromycin could not be considered a cause of concern, since in the standardised methods for the determination of the MIC, a variability of one doubling dilution of the real end point is generally accepted [13,19,46]. Furthermore, in order to determine which species and antibiotic resistance genes were present, molecular analysis on DNA extracted from cultures grown in LSM and ISTL broth media supplemented with erythromycin (2, 4, or 8 mg/mL) or tetracycline (4, 8, or 16 mg/mL) and from colonies derived from them revealed the only presence of the *E. faecium* species. The *tet*M was the only resistance gene detected in 30 colonies, coming from tetracycline-supplemented broth, out of the 49 investigated (data not shown).

Among the species detected in SR30, SR56 and SR63, *E. faecium* is the only one not included in the EFSA Qualified Presumption of Safety (QPS) list. The QPS concept was introduced by the European Food and Safety Authority (EFSA) in 2007. All bacterial strains belonging to species for which there is a sufficient body of knowledge and a long history of safe use can be considered safe and included in the QPS list drawn up by BIOHAZ [47]. The requests of indication for assessing the safety of *E. faecium* as feed additive or as production organisms have increased in recent decades [22], since about one third of the microbial additives authorised by EFSA contains *E. faecium* strains. To be considered safe, the strains must show ampicillin MIC ≤ 2 mg/L and an absence of clinical importance markers (*hyl*Efm, *esp*, and IS*16*) [13]. The investigation of 41 *E. faecium*, representing each biotype detected, revealed no ampicillin resistance, since all the strains were unable to grow at ampicillin 0.5 mg/L, while two showed a MIC = 1 mg/L. Furthermore, no pathogenic factors *hyl*Efm, *esp*, and IS*16* were detected in any strains investigated [13]. Ampicillin MIC ≤ 2 mg/L and the absence of virulence factors such as *hyl*Efm (coding for a glycosyl hydrolase involved in the colonisation of gastrointestinal tract) [43], *esp* (part of a pathogenicity island and coding for a surface protein important for biofilm formation) [48], and IS*16* (transposable element and hospital-associated marker) [49], allow the use of *E. faecium* strains as feed additive or as production organisms [13]. This screening allows the discrimination between livestock commensal and food-associated *E. faecium* and multiresistant hospital-associated ones [49,50], reducing the risk of possible horizontal gene transfer among different reservoirs [51]. Indeed, the phenotypic resistance reflects the presence of a biochemical mechanism and a relative genetic determinant of resistance evolved at the species level. If this resistance, for a certain antibiotic, is expressed by most strains of a bacterial species, the species is considered to be endowed with intrinsic and not transferable resistance [22,52]. However, among the strains belonging to a sensitive species, resistant strains may emerge due to a recent genetic modification in the genome, inducing acquired resistance. The genetic bases of this phenomenon include mutation or recombination of resident genes or acquisition of new genes by horizontal gene exchange [48,53,54]. This may cause clinical risk in case LAB, present in food, transfer resistance to pathogenic bacteria in food or in the gastrointestinal tract of humans and animals [55,56,57]. The main results are summarised in Table 3.

At present, there is no legislation requiring the control of the presence of antibiotic resistance in natural cultures used as starters for cheese manufacturing. However, as a precautionary measure, with a view of a possible tightening-up of the legislation in this regard, and in order to be able to certify the absence of resistant antibiotic bacteria as an added value on the product label, this aspect was investigated as well.

## 4. Conclusions

In this study, the composition and safety of natural starter cultures, preserved *in toto* in freeze-dried form since the late 1960s were investigated. The biodiversity rate found at strain level, though high, could have been underestimated since only the dominating species/strains able to grow in the laboratory conditions (i.e., through culture-dependent techniques) were taken into account, while the use of more powerful culture-independent techniques could help to deepen knowledge regarding all the microbial communities that make up the cultures. Moreover, the encouraging results obtained regarding the antibiotic resistance of the *scotta-innesto* microflora investigated, and the compliance of *E. faecium* isolates with the EFSA indications for QPS, support the possibility of safely using the SR30, SR56, and SR63 natural cultures to produce Pecorino Romano PDO, avoiding the use of commercial starters. Moreover, the production of natural biodiverse cultures, at large scale, in freeze-dried form, could overcome the problems associated with the daily propagation of a natural *scotta-innesto*, whose technological efficiency is not always guaranteed, at the dairy plant. The use of natural cultures could have positive repercussions in terms of sensory richness, peculiarity, and uniqueness of the final product. These are very important aspects for artisanal, typical and PDO cheeses, which, thanks to the raw material to be processed—the autochthony and biodiversity of the microbial cultures—would be more linked to the territory of production.

All the cultures and the isolates investigated in the present study, characterised at the strain-level and for virulence factors and antibiotic resistance profiles, are stored at −80 °C and maintained at the BNSS Microbial Collection, associated member of the Joint Research Unit (JTU) MITTI-IT (Microbial Resource Research Infrastructure Italian Node), located at Agris Sardegna, Italy, for further characterisation studies.

## Figures and Tables

**Figure 1 microorganisms-09-01363-f001:**
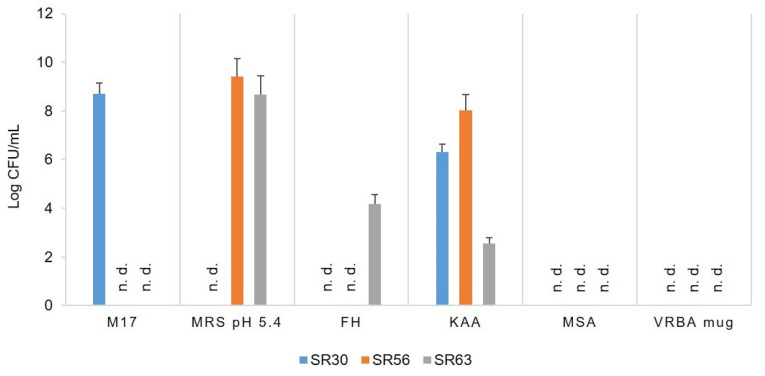
Microbial counts of thermophilic cocci (M17), thermophilic lactobacilli (MRS pH 5.4), mesophilic lactobacilli (FH), enterococci (KAA), staphylococci (MSA), and coliforms (VRBA mug) in the natural starter cultures SR30, SR56, and SR63. For each microbial group, counts are expressed as Log CFU/mL ± standard deviation (SD). n.d., not detected.

**Figure 2 microorganisms-09-01363-f002:**
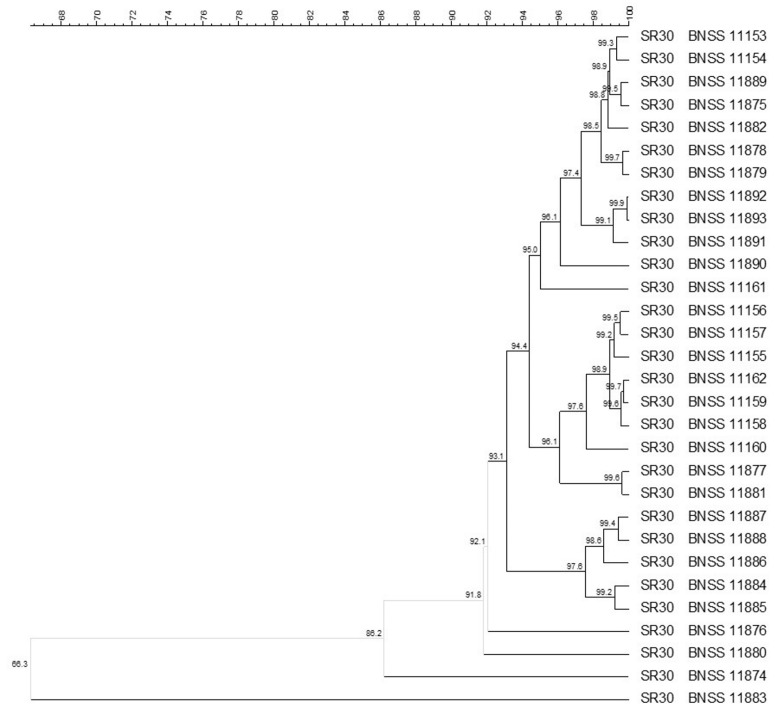
UPGMA cluster analysis, using Pearson correlation, of (GTG)_5_ rep-PCR and PFGE elaborated by composite data set comparison tool, of 30 *Streptococcus thermophilus* isolates from SR30 natural culture. For each isolate, the indication of the culture and the BNSS ID were reported.

**Figure 3 microorganisms-09-01363-f003:**
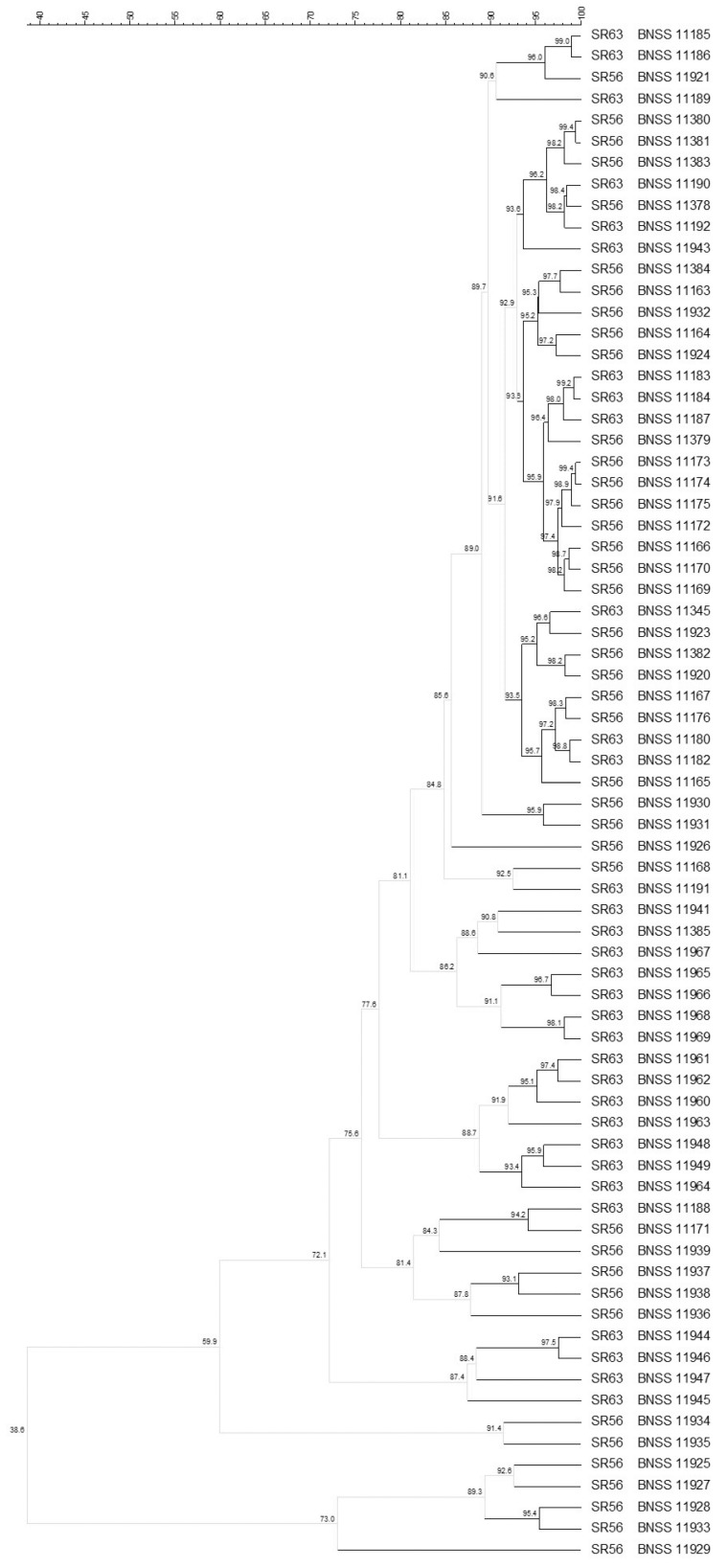
UPGMA cluster analysis, using Pearson correlation, of (GTG)_5_ rep-PCR and PFGE elaborated by composite data set comparison tool, of 72 *Lactobacillus delbrueckii* subsp. *lactis* isolates from SR56 and SR63 natural cultures. For each isolate, the indication of the culture (56 or 63) and the BNSS ID were reported.

**Figure 4 microorganisms-09-01363-f004:**
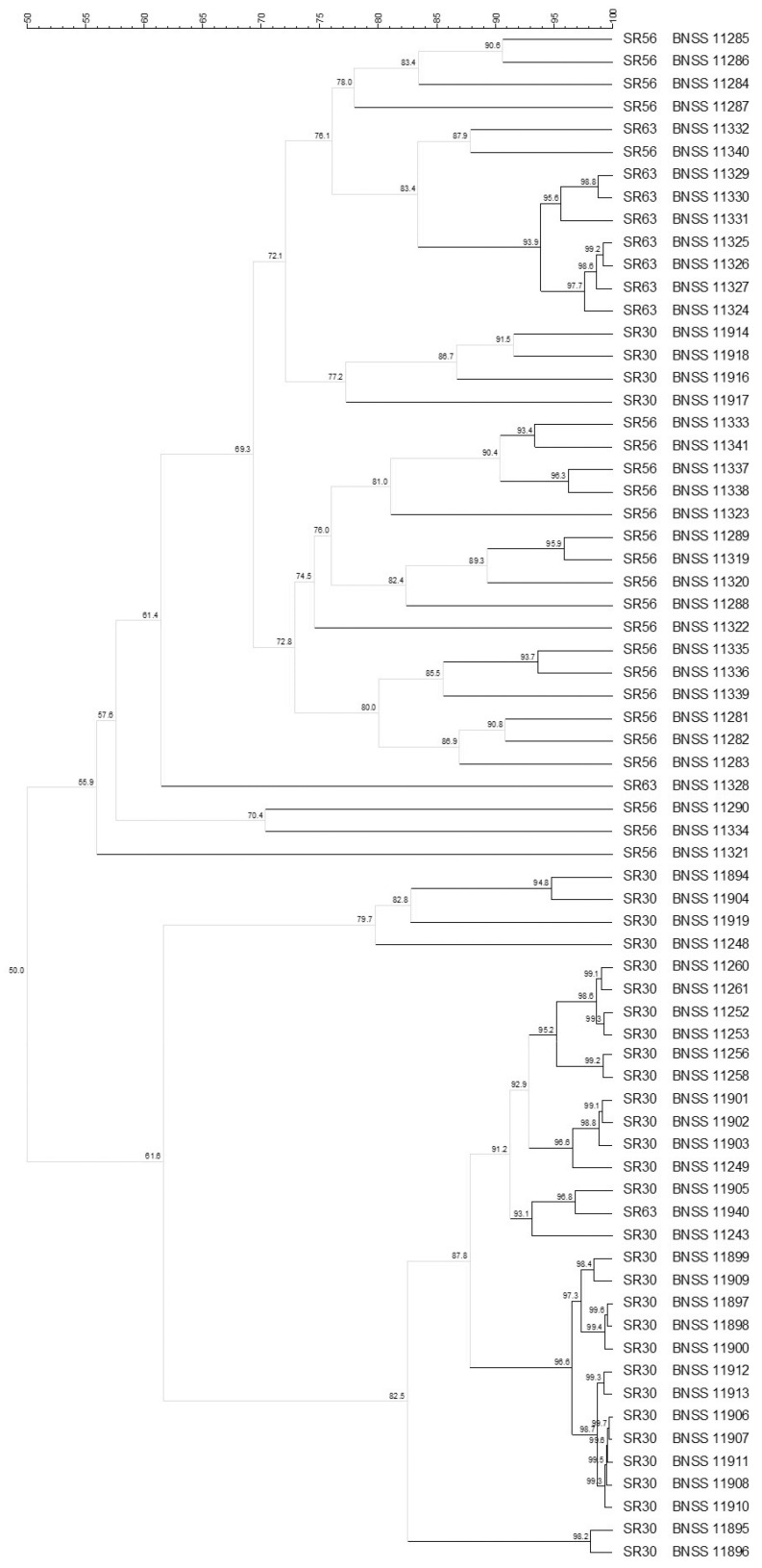
UPGMA cluster analysis, using Pearson correlation, of (GTG)_5_ rep-PCR and PFGE elaborated by composite data set comparison tool, of 68 *Enterococcus faecium* isolates from SR30, SR56, and SR63 natural cultures. For each isolate, the indication of the culture (30, 56 or 63) and the BNSS ID were reported.

**Figure 5 microorganisms-09-01363-f005:**
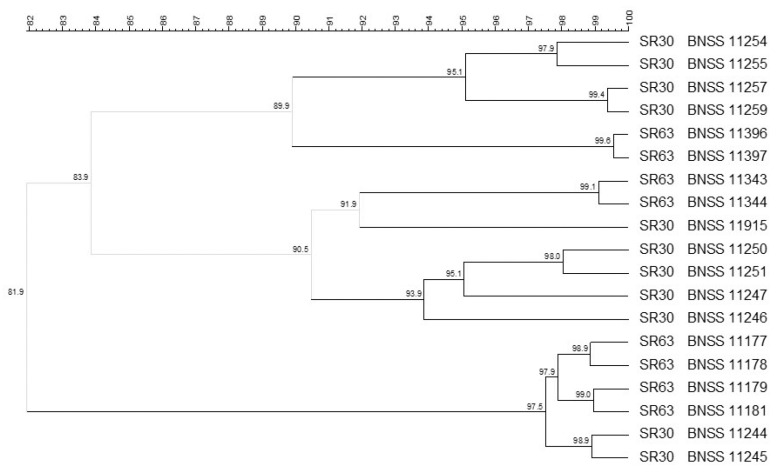
UPGMA cluster analysis, using Pearson correlation, of (GTG)_5_ rep-PCR fingerprints of 19 *Enterococcus faecium* isolates from SR30 and SR63 natural cultures. For each isolate, the indication of the culture (30 or 63) and the BNSS ID were reported.

**Figure 6 microorganisms-09-01363-f006:**
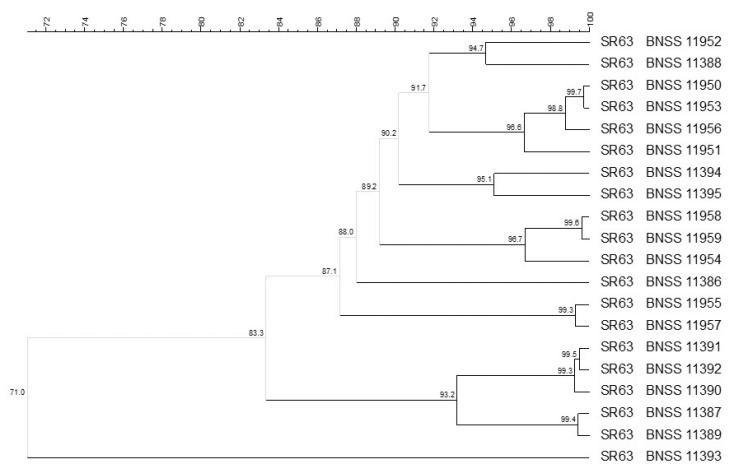
UPGMA cluster analysis, using Pearson correlation, of (GTG)_5_ rep-PCR and PFGE elaborated by composite data set comparison tool, of 20 *Limosilactobacillus reuteri* isolates from SR63 natural cultures For each isolate, the indication of the culture and the BNSS ID were reported.

**Table 1 microorganisms-09-01363-t001:** Evaluation of antibiotic resistance of the SR30, SR56, and SR63 cultures *in toto*, in media supplemented with different antibiotic concentrations.

Antibiotic	Concentration	Culture Medium	Microbial Culture		
	(mg/L)		SR30	SR56	SR63
Amp	2 ^1^	LSM	−	−	−
		ISTL	n.t.	n.t.	n.t.
	4	LSM	−	−	−
		ISTL	−	−	−
	8 ^2^	LSM	−	−	−
		ISTL	n.t.	n.t.	n.t.
Chl	8 ^1,2^	LSM	−	−	−
		ISTL	−	−	−
	16	LSM	−	−	−
		ISTL	n.t.	n.t.	n.t.
	32	LSM	n.t.	n.t.	n.t.
		ISTL	−	−	−
Cli ^2^	2	LSM	+	+	−
		ISTL	n.t.	n.t.	n.t.
	4 ^1^	LSM	n.t.	n.t.	n.t.
		ISTL	+	+	−
	8	LSM	n.t.	n.t.	n.t.
		ISTL	−	+	−
Ery	1	LSM	+	+	−
		ISTL	+	+	−
	2	LSM	−	+	−
		ISTL	−	+	−
	4 ^1,2^	LSM	−	+	−
		ISTL	−	+	−
	8	LSM	−	+	−
		ISTL	−	+	−
Gen	16	LSM	−	+	+
		ISTL	n.t.	n.t.	n.t.
	32 ^1^	LSM	−	+	−
		ISTL	−	−	−
Tet	4 ^1^	LSM	−	+	−
		ISTL	−	+	−
	8 ^2^	LSM	−	+	−
		ISTL	−	+	−
	16	LSM	−	+	−
		ISTL	−	−	−
Til	8 ^3^	LSM	−	−	−
		ISTL	−	−	−

ISTL, IST medium for cocci supplemented with 10 g/L lactose; LSM, medium for lactobacilli; Amp, ampicillin; Chl, chloramphenicol; Cli, clindamycin; Ery, erythromycin; Gen, gentamycin; Tet, tetracycline; Tyl, tylosin. The antibiotic concentrations flanked by apex numbers are referred to the cut-offs indicated for *E. faecium* by: ^1^ EFSA Panel on Additives Products or Substances used in Animal Feed 2018; ^2^ CLSI M100-S30 (2020) Table 2D; ^3^ NARMS 2016/2017; n.t., not tested.

**Table 2 microorganisms-09-01363-t002:** Estimation of tetracycline and erythromycin resistance bacteria in SR56 culture.

Antibiotic	Concentration	Culture Medium	Microbial Dilution ^1^		
	(mg/L)		1/10	1/100	1/1000
Ery	2	LSM	+	+	+
		ISTL	+	−	−
	4	LSM	+	+	+
		ISTL	+	−	−
	8	LSM	+	+	−
		ISTL	+	−	−
Tet	4	LSM	+	+	+
		ISTL	+	+	+
	8	LSM	+	+	+
		ISTL	+	+	+
	16	LSM	−	−	−
		ISTL	−	−	−

ISTL, IST medium for cocci supplemented with 10 g/L lactose; LSM, medium for lactobacilli; Ery, erythromycin; Tet, tetracycline. ^1^ A volume of 0.1 mL was inoculated in each media, the results must be referred to the next higher dilution.

**Table 3 microorganisms-09-01363-t003:** Summary of the main results.

	**Biodiversity**				**Safety Assessment ^1^**							
**Cultures**	**Species**	**Log CFU/mL**	**Number of Isolates**	**Number of Biotypes**	**Phenotypic Antibiotic Susceptibility ^2^**		**Molecular Antibiotic Resistance ^3,4^**		***E. faecium* Isolates ^5^ Specific Tests [13]**			
					**Cultures *in toto***	**CFU/mL Estimation**	**Cultures *in toto***	**Colonies**	**Amp Resistance**	***esp***	***hyl*** **_efm_**	**IS*16***
SR30					Amp Neg Chl Neg Cli 4 mg/L Ery 1 mg/L Gen Neg Tet Neg Til Neg	n.t.	n.t.	n.t.	<2 mg/L	Neg	Neg	Neg
	*S. thermophilus*	8.72	30	5								
	*E. faecium*	6.32	45	16								

SR56					Amp Neg Chl Neg Cli 8 mg/L Ery 8 mg/L Gen 32 mg/L Tet 16 mg/L Til Neg	Ery_2/4 mg/L_: 4 Log Ery_8 mg/L_: 3 Log Tet_4/8 mg/L_: 4 Log	*tetM*	*tetM*	<2 mg/L	Neg	Neg	Neg
	*E. faecium*	8.03	24	20								
	*L. delbrueckii lactis*	9.41	40	17								

SR63					Amp Neg Chl Neg Cli Neg Ery Neg Gen 16 mg/L Tet Neg Til Neg	n.t.	n.t.	n.t.	<2 mg/L	Neg	Neg	Neg
	*E. faecium*	2.56	18	7								
	*L. delbrueckii lactis*	8.67	32	18								
	*L. reuteri*	4.18	20	8								


^1^ in ISTL and LSM media supplemented with Amp, Chl, Cli, Ery, Gen, Tet, and Til at different concentrations (see Table 1); ^2^ refers to the highest antibiotic concentration with positive microbial growth and was due only to *E. faecium*; ^3^ *E. faecium* was the only species detected both in the SR56 culture *in toto* and among 49 colonies; ^4^ antibiotic resistance genes tested: *tet*M, *tet*L, *tet*S, *tet*W, *tet*K, *erm*A, *erm*B. *tet*M was the only gene detected in 30 out of 49 colonies; ^5^ one representative for each of the *E. faecium* biotypes from the three cultures; n.t., not tested.

## Data Availability

Not applicable.

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
