# Peer review of "Biodiversity and Safety Assessment of Half-Century Preserved Natural Starter Cultures for Pecorino Romano PDO Cheese"

_microorganisms, 2021, doi:10.3390/microorganisms9071363_

Round 1

Reviewer 1 Report

The manuscript describes the biodiversity and safety assessment of half-century preserved natural starter cultures for the manufacture of the Italian PDO cheese Pecorino Romano. I found it quite interesting and very pleasant to read and I therefore recommend its publication. 

The authors addressed the subject scientifically well and I believe the findings would be of significant importance to the Pecorino Romano industry. 

My only comment would be to include in the introduction (i) some numerical data regarding the quantities of the cheese produced and exported, (ii) is there are differences reported in the microbiome of cheese made from raw Vs thermised milk and (iii) if they are more and how many autochthonous natural scotta-innesto.

Author Response

The authors would like to thank the Reviewers for the positive and useful comments and the suggestions that will help to improve the level of the manuscript.

It should be noted that, since the first submission, the molecular analyses regarding the characterization of L. reuteri at strain level were completed. Therefore, the results reported in the revised manuscript version were updated accordingly.

All changes on the manuscript were performed using the MS Word Track Changes tool.

The answers to the Reviewers’ comments are listed below.

Answers to the Reviewers’ 1 comments

Question: “include in the introduction (i) some numerical data regarding the quantities of the cheese produced and exported”

Answer: data were added in the Introduction section. The amount of Pecorino Romano PDO exported in the world is about 20% (https://www.pecorinoromano.com/application/files/8415/7323/4642/Pecorino_romano_DOP_Gennaio_Ago_2019_Ismea.pdf) of the total Pecorino cheese https://www.clal.it/index.php?section=imp_exp_istat&cod=04069063&mov=E)

Question: “include in the introduction (ii) is there are differences reported in the microbiome of cheese made from raw Vs thermised milk”

Answer: The Product specification establishes that the milk may undergo to thermisation (EC No. 1030/2009). However, nowadays no raw milk is normally used for Pecorino Romano cheesemaking, therefore no data are available. The sentence at L. 30 of the submitted manuscript was slightly modified.

Question: “include in the introduction and (iii) if they are more and how many autochthonous natural scotta-innesto”

Answer: The Agris Sardegna BNSS microbial collection (http://www.mbds.it) includes about 80 more natural scotta-innesto for Pecorino Romano. We just have started by investigating 3 of these cultures but in the next future we will likely reactivate and investigate also other cultures stored in this collection.

Reviewer 2 Report

An interesting good paper which can be improved.

Author Response

The authors would like to thank the Reviewers for the positive and useful comments and the suggestions that will help to improve the level of the manuscript.

It should be noted that, since the first submission, the molecular analyses regarding the characterization of L. reuteri at strain level were completed. Therefore, the results reported in the revised manuscript version were updated accordingly.

All changes on the manuscript were performed using the MS Word Track Changes tool.

The answers to the Reviewers’ comments are listed below.

Answers to the Reviewers’ 2 comments

Question: please add a compilation table for the data you found for these three SR samples.

Answer: A Summary Supplementary Table S2 including the main data obtained about the cultures’ characterization was added to the revised manuscript. The authors believe that the addition of this table in the main text should be redundant.

Question: From the data put in this table is it plausible to suggest a higher usefullness for one or another scotta innesta beyond the three studied here ???? For SR 56 it seems to contain a lot of resistance to some antibiotics compared to the two other SR although it does not change anything for the QPS status . However from a safety point of view is it your policy to use such SR 56 while it seems you have two others which have different characteristics? It will be better to be careful in this field of resistance to antibiotics or not ??? How do you explain such difference of SR 56 compared to the two other SR???are these antibiotics possibly present in some ewe’s milk to explain such a selection of antibiotic resistance in SR56 bacterial content in toto????

Answer: We do not have information about antibiotic administration to ewes during the years of the cultures sampling from the dairy farms, in the 1968. Although the SR56 revealed higher antibiotic resistance compared to the other two cultures, it was below the cut-offs suggested by different institutions for E. faecium (the only species able to grow in the antibiotic supplemented media used). It is noteworthy that the antibiotic resistance genes in the SR56 culture and its isolates were detected only stressing the test using antibiotic-supplemented media. Moreover, in a next step of this study we investigated the antibiotic resistance also in the cheeses obtained using these cultures, and the level of resistant bacteria was very low and near to the detection level, so cheeses can be consumed safely. We did not put this information in the present manuscript because these data will be a part of a future manuscript following the present one and the two recently published (Chessa et al., 2019 and 2020).

Question: Additionally In this Summary Table you could report also about the interest for use in pecorino production for one or another of these three SR samples.

Answer: The main purpose of this study was the characterization of three natural cultures, candidate for a future use in cheesemaking. The three natural cultures characterized are different for their microbial composition in starter lactic acid bacteria. Indeed, SR30 contains only S. thermophilus, while SR56 and SR63 are constituted mainly by thermophilic lactobacilli but lacking in thermophilic cocci. Therefore, they should be combined in order to have good technological performances. As mentioned before, specific trials regarding the use of the SR cultures in cheesemaking will be widely described in a new paper that is going to be submitted shortly.

Question: Is there any possibility to visualize in a concrete way by putting a comparison of your SR with some commercial starters (lines 199 to 232 you did a good explanation which could be illustrated or not ???)?? So at the end of your draft you should conclude to this in what way you have obtained a better lactic acid microflora than that present in the commercial starter cultures.

Answer: A comparison of the microbiota of scotta-innesto and cheeses obtained using natural or commercial starter was not carried out in this study, but it is the aim of the manuscript mentioned above. We would like to thank the Reviewer for this question because it means that maybe that manuscript could meet the interests of the scientific community operating in this field.

Question: I see that your main issue is for a “slow food approach” to ensure safety and specificity through the use of autochtonous starters and you wrote a balanced text in the conclusion: “Moreover, the encouraging results obtained regarding antibiotic resistance of the scotta-innesto microflora investigated, and the compliance of E. faecium isolates with the EFSA indications for QPS, support the possibility to use safely the SR30, SR56, and SR63 natural cultures to produce Pecorino Romano PDO avoiding the use of commercial starters. Moreover, the production of natural biodiverse cultures, at large scale, in freeze-dried form, could overcome the problems associated with the daily propagation of a natural scotta-innesto, whose technological efficiency is not always guaranteed, at the dairy plant. The use of natural cultures could have positive repercussions in terms of sensory richness, peculiarity, and uniqueness of the final product. These are very remarkable aspects for artisanal, typical and PDO cheeses, that, thanks to the raw material to be processed, the autochthony and biodiversity of the microbial cultures, would be 420 more linked to the territory of production ». “ and this was summarized in the abstract as “The use of natural biodiverse cultures for the production of microbial starters for typical and PDO cheeses, such as Pecorino Romano, could be an opportunity for recovering the cheese microbiota biodiversity lost during years of commercial starters use.” What is your postion about Slow food movement???

Answer: Despite we did not consider the principles inspiring Slow food in designing and performing this study, the protection and enhancement of microbial biodiversity, which is one of the institutional purposes and expertise of AGRIS Sardegna are in line with one of Slow food main purposes and contribute to prevent the disappearance of local food cultures and traditions.

Question: I would like to thank you very much and congratulations! It is not easy to counterbalance the European policy which had sometimes failed to success due to lost of the “soul” of a traditional food by using an excess of safety criteria especially at the period of the beginning of such policy ….

Answer: The authors thank the Reviewer for the positive comments about our work. We totally agree since traditional and typical food products needs to regain the right space in the European context.

Specific Reviewer comments

Question: I find that the title of the present draft is in agreement with the content of the draft but not with the content of your Abstract since in it safety point was quoted first in place of Biodiversity study ….

Answer: The Abstract was modified according to the reviewer’s suggestions.

Question: Line 46 ref 9 should be numbered 10 and ref 10 should be numbered 9 to comply with date of publication and keep it in time order.

Answer: We cannot choose or change the references numbers. We used a bibliography software package furnished by the Journal to handle the citations.

Miscellaneous Reviewer comments

Question: Ligne 163: “PCR for the identification at genus (Enteroccoccus spp.) or species (E. faecium, E. faecalis, E. durans, and S. thermophilus)” Have you found E durans ??? E faecalis??

Answer: No, we detected only E. faecium, as reported in L. 360-361 of the submitted manuscript version “from colonies derived from them revealed the only presence of E. faecium species”.

Question: Ligne 82 : Bacilli → Lactobacilli better ??

Answer: Improved as suggested.

Question: Ligne 243 : the indication of the culture is SR30 and not only 30 …in fact 30 is confusing so remove from each species identified and put only in the legend with aid of *

Answer: the Figure legend was improved to avoid misunderstanding. Moreover, the Figures, from 2 to 6, were improved adding the correct indication of the culture: SR30, SR56, and SR63

Question: Line 231: genera Staphylococcus in italic please

Answer: Done.

Question: Figure 1: what is the meaning of n.d.???

Answer: n.d. means “not detected”. It was added in the Figure 1 caption of the revised manuscript.

Question: Figure 3 put * or ** for 56 or 63 which are confusing numbers

Answer: The Figures were modified as stated above.

Question: Lignes243 à 244 : The molecular biotyping performed on 72 L. delbrueckii subsp. lactis isolates revealed 30 different biotypes isolated from SR56 and SR63 natural starter cultures (Figure 3). In particular, 54 of the L. delbrueckii subsp. lactis isolates were distributed into 12 clusters, plus 18 singletons . Please put the cut off (< to ????) to justify this sentence = 93% similarity cut-off

Answer: The sentence was improved as suggested.

Question: The writing : n. in your text is for you an abbreviation of number ! please put in another form to clarify since it concerns the number of isolates (example line 255 and before line 201 and 202 and lines 258, 259)

Answer: “n.” was replaced with “No.”

Question: Line 257 : 93% similarity cut-off to arrive to 35 different biotypes????

Answer: The cut-off was added.

Question: Figure 4 line 272-273 remove confusing numbers in your dendogram and put * or ** or *** for 30, 56 and 63 respectively for the identification of the natural starter they come from

Answer: The indication of the cultures (SR30, SR56, and SR63) was added in the figures.

Question: Figure 5 and 6 idem same improvements are asked like for previous figures

Answer: Improved like Figure 4.

Round 2

Reviewer 2 Report

See enclosed file 

Author Response

Answers to the Reviewer comments

Question: →In Title : preliminary approach to safety assessment ….Presently to be added in the title . Furthermore you cannot use as a keyword Qualified Presumption of Safety (QPS) since it is a generic approach done by EFSA…

Answer: The authors consider the approach used not preliminary since the sensitivity to several antibiotics was tested, and the safety assessment for non-QPS species was performed, as indicated by the EFSA, therefore they would prefer to keep the original title of the manuscript.

QPS was removed from the keywords.

Question: →I was surprised by your answer: “A Summary Supplementary Table S2 including the main data obtained about the cultures’ characterization was added to the revised manuscript. The authors believe that the addition of this table in the main text should be redundant”. I don’t agree with that , at the contrary it allows better understanding so you must put this Table at the right place in the text (and not in Supplementary material) since it clarifies the lot of enumeration of species and of results difficult to digest especially for the comparison between in toto and with isolates …..

Answer: The Supplementary Table S2, summarising some of the main results obtained, was moved into the main text as Table 3, as requested.

Question: Futhermore the use in your draft V1 of n. replace now by No is very confusing since we don’t know if it a number designing one isolate or the number of isolates studied ….You must be more clearer…

Answer: In the text and in Figure 2 caption, in our opinion it was clear that “n.” or “No.” refer to the number of isolates investigated and not to a particular isolate or strain (in UPGMA figures each strain is indicated with its univocal BNSS ID).  In the last revised manuscript version, all the “n.” were replaced with “No.” using the MS Word track changes tool. In any case, to avoid confusion, the same way to indicate the numbers of the isolates tested was used in whole manuscript, deleting all the “No.”.

Question: However I understand this answer since it shows that a lot of results are missing and it is difficult to understand the common thread???

Answer: Actually, all the results fitting with the aim of this study were presented.

Question: → Another answer from you is also very surprising as you are already writing another paper and forget the present one when I ask you a lot of supplementary details or comments you systematically tell me that it is for the other draft ! I hope you know that normally you are not authorized to write another paper similar to the present one

Answer: The authors are fully aware that is not possible to publish the same data in different papers, and consider the Reviewer’s statement quite offensive of their intellectual honesty. Moreover, the authors confirm, as stated in the previous point-by-point rebuttal letter, that this manuscript is a part of a larger project investigating the Pecorino Romano PDO cheese, as reported also by the Reviewer in the previous revision.

Indeed, the Reviewer wrote “The present study done and written by… is a logical continuation of a project sustained by Agris… and for which two basic papers has been already published by the same team… on this subject”.

In brief:

  1. in 2019, the same authors investigated the “Effect of growth media on natural starter culture composition and performance evaluated with a polyphasic approach”;
  2. in 2020 the same cultures were investigated for the “Optimization of scotta as growth medium to preserve biodiversity and maximise bacterial cells concentration of natural starter cultures for Pecorino Romano PDO cheese”.
  3. in the present manuscript, the cultures were investigated to assess their biodiversity, at strain-level, and safety.
  4. The next step will be to write a paper concerning the use of these cultures in Pecorino Romano cheesemaking. Therefore, the comparison among these cultures and between natural and commercial cultures in cheesemaking, asked by the Reviewer in the last revision, will be the matter of a new manuscript, where new data will be presented.

Question: …at the same time: it is about your answer “The main purpose of this study was the characterization of three natural cultures, candidate for a future use in cheesemaking. The three natural cultures characterized are different for their microbial composition in starter lactic acid bacteria. Indeed, SR30 contains only S. thermophilus, while SR56 and SR63 are constituted mainly by thermophilic lactobacilli but lacking in thermophilic cocci. Therefore, they should be combined in order to have good technological performances. As mentioned before, specific trials regarding the use of the SR cultures in cheesemaking will be widely described in a new paper that is going to be submitted shortly.

My comments : How do you assert that SR 30 contains only S thermophilus while you say the contrary in your draft (→A total of 209 isolates from the natural cultures were molecularly identified at species level: 30 S. thermophilus and 45 E. faecium were isolated from SR30. Both cocci-shaped bacteria (E. faecium, Non. 24 and Non. 18) and homofermentative thermophilic lactobacilli (L delbrueckii subsp. lactis, n.No. 40 and n.No. 32) were isolated from SR56 and SR63, respectively, and 20 Limosilactobacillus reuteri were isolated from SR63

Answer: In the last rebuttal letter, the authors affirmed: "The ...cultures...are different for their microbial composition in starter lactic acid bacteria" considering only the species able to lead the early acidifying fermentation phase while in the text the complete microbial composition was reported highlighting differences in starter (SLAB) and non-starter LAB (NSLAB) (lines 205-238 of the manuscript version revised R2). Therefore, no contradiction was in the statements.

Question: →I see that you change the title of Table 1 which was first : “Evaluation of antibiotic resistance of the (microbial isolates, isolated from) SR30, SR56, and SR63, cultures, in toto, in media supplemented with different antibiotic concentrations.” So maybe you understand why I ask you to summarize the results in the text by adding a new Table in order to know what have been really done ….. Furthermore it means you have first in the V1 presented results with a false Caption….

Answer: Though antibiotic sensitivity was investigated both for cultures in toto and several isolates coming exclusively from SR56 culture, Table 1 contains only data related to cultures in toto, whereas data concerning isolates were mentioned in the text (lines 364-369 manuscript revised R2). The title of the Table 1 was amended accordingly.

Question: → About my question: Is there any possibility to visualize in a concrete way by putting a comparison of your SR with some commercial starters (lines 199 to 232 you did a good explanation which could be illustrated or not ???)?? So at the end of your draft you should conclude to this in what way you have obtained a better lactic acid microflora than that present in the commercial starter cultures. And your Answer: A comparison of the microbiota of scotta-innesto and cheeses obtained using natural or commercial starter was not carried out in this study, but it is the aim of the manuscript mentioned above. We would like to thank the Reviewer for this question because it means that maybe that manuscript could meet the interests of the scientific community operating in this field. YES I ASK you to do that by using the known content of some commercial starter even if you don’t check the efficiency of these commercial starters . I ask to tell us what are these commercial starters composed of??? Some words are sufficient…

Answer: About the question “So at the end of your draft you should conclude to this in what way you have obtained a better lactic acid microflora than that present in the commercial starter cultures” the authors confirm that the comparison between commercial and natural starters is not the focus of this manuscript. Therefore, the data presented cannot support the reviewer’s statement since no commercial starters were investigated in this study. However, in the introduction the sentence regarding commercial starters was improved as follows: “… commercial starter cultures consisting of few selected strains of lactic acid bacteria (LAB), such as Streptococcus thermophilus, Lactobacillus delbrueckii, Lactobacillus helveticus, certified as autochthonous.” (lines 42-44 of the manuscript version revised R2).

Question: To conclude: So sorry it is sufficient for me! you have to improve the quality of your writing and the background too …in order to write a perfect draft which can be without such mistakes you did (see above) and which can be by any competent reader of the journal Microorganisms…the main fact is to use the new table S2 as a guide and to fill all the gaps with corresponding data ! you did a big and good job about identification of the species but there is too much discrepancy with the reduce data presented for the other experiments. You could represent this paper after having fill gaps…

Answer: As suggested, Table S2 was moved into the main text as Table 3, in order to better summarise the main results obtained. Furthermore, the authors hope to have explained their position about the gaps identified by the Reviewer, and to have convinced her/him that the huge work presented in this article does not allow to add even the results of the cheesemaking trials, which deserve a dedicated article.

Round 3

Reviewer 2 Report

Specific comments :

--> Check the n. and No (Table 3) and suppress please .

--> In your previous answer: you quoted :  “Question: However I understand this answer since it shows that a lot of results are missing and it is difficult to understand the common thread???

Your Answer was : Actually, all the results fitting with the aim of this study were presented

So my present comments are then about this point…. And thus it concerns the table 3 where there is a lot of n.t (not tested ) . If it corresponds to E faecium isolates I can understand but in that case you must put it is isolates. So could you explain to me why you haven’t tested Amp sensitivity for SR 30 in toto and SR 63 in toto??? The must is that you did in Table 1 so could you harmonized the content of this table 3 to the one of table 1 in accordance to the title of your draft about safety assessment i.e presence or absence of Amp gene in E faecium and /or in toto  …. The results in toto dealing with Amp results does not appear (summarized) in any part of your table 3 for SR 30 or SR 63 !  Put Amp results even if negative !

So only when you do this clarification and present a right table 3 I could accept your  very valuable paper…

Author Response

Answers to the Reviewer’s questions

Question: --> Check the n. and No (Table 3) and suppress please .

Answer: all the n. and No were removed from Table 3.

Question: --> In your previous answer: you quoted :  “Question: However I understand this answer since it shows that a lot of results are missing and it is difficult to understand the common thread???

Your Answer was : Actually, all the results fitting with the aim of this study were presented

So my present comments are then about this point…. And thus it concerns the table 3 where there is a lot of n.t (not tested ) . If it corresponds to E faecium isolates I can understand but in that case you must put it is isolates. So could you explain to me why you haven’t tested Amp sensitivity for SR 30 in toto and SR 63 in toto??? The must is that you did in Table 1 so could you harmonized the content of this table 3 to the one of table 1 in accordance to the title of your draft about safety assessment i.e presence or absence of Amp gene in E faecium and /or in toto  …. The results in toto dealing with Amp results does not appear (summarized) in any part of your table 3 for SR 30 or SR 63 !  Put Amp results even if negative !

So only when you do this clarification and present a right table 3 I could accept your  very valuable paper…

Answer: Table 3 was improved as suggested, and “isolates” was added to the heading.

As reported in Table 1, all the three cultures in toto were tested for ampicillin sensitivity. As suggested by the Reviewer, to better summarise these results, the absence of ampicillin resistance was indicated also in Table 3.

Moreover, though the ampicillin sensitivity and the evaluation of virulence genes were assessed also for the E. faecium isolates representative of each of the biotypes detected in the three SR cultures analysed (as reported in lines 384-387 of the manuscript), in Table 3 we erroneously typed n.t. instead of the results obtained.

Furthermore, during the last revision we noticed that the 6 E. faecium isolates, representative of the 6 biotypes detected among the 19 isolates typed only with (GTG)5 (Fig. 5), were not reported in the total number of isolates tested for Amp susceptibility and presence of the three virulence genes (line 384).

Therefore, data were updated in this manuscript_R3 version (line 384) and Table 3 was amended.

Round 4

Reviewer 2 Report

Ok it is good now and the authors has  recognized

They erreounously did mistakes....and they  change 

the data in Table 3 ! ?